

# Atmospheric controls on hydrogen and oxygen isotope composition of meteoric and surface waters in Patagonia

Christoph Mayr[1,2], Lukas Langhamer[3], Holger Wissel[4], Wolfgang Meier[1], Tobias Sauter[1], Cecilia Laprida[5], Julieta Massaferro[6], Günter Försterra[7], Andreas Lücke[4]

[1]Institute of Geography, Friedrich-Alexander-Universität Erlangen-Nürnberg, Erlangen, 91058, Germany
[2]GeoBio-Center and Department of Earth and Environmental Sciences, Ludwig-Maximilians-Universität München, München, 80333, Germany
[3]Institute of Atmospheric and Cryospheric Sciences, Universität Innsbruck, Innsbruck, 6020, Austria
[4]Institute of Bio and Geosciences, IBG-3: Agrosphere, Forschungszentrum Jülich, Jülich, 52425, Germany
[5]Instituto de Estudios Andinos "Don Pablo Groeber", Universidad de Buenos Aires – CONICET, Buenos Aires, C1428EHA, Argentina
[6]CONICET, CENAC/APN, Bariloche, 8400, Argentina
[7]Huinay Scientific Field Station, and Escuela de Ciencias del Mar, Facultad de Recursos Naturales, Pontificia Universidad Católica de Valparaíso, Valparaíso, Chile

*Correspondence to*: Christoph Mayr (christoph.mayr@fau.de)

**Abstract.** The southern tip of South America, commonly referred to as Patagonia, is a key area to understand Southern Hemisphere Westerlies (SHW) dynamics and orographic isotope effects in precipitation. However, only few studies have addressed these topics. We evaluated the stable isotope ($\delta^2$H, $\delta^{18}$O) compositions of precipitation, lentic waters, and lotic waters in that area to characterize and understand isotope fractionation processes associated with orographic rainout, moisture recycling and moisture sources. Observational data were interpreted with the help of backward trajectory modelling of moisture sources using reanalysis climate data. While the Pacific serves as the exclusive moisture source for sites upwind of the Andes and on the immediate downwind area of the Andes, recycled moisture from the continent seems to be the main humidity source at the Patagonian Atlantic coast. In contrast, the Pampean Atlantic coast north of Patagonia obtains moisture from the Atlantic Ocean. In the core zone of the SHW at a latitude of 50° S, a depletion in the heavy isotopes of 10 ‰ and 85 ‰, for $\delta^{18}$O and $\delta^2$H, respectively, occurs due to orographic rainout corresponding to a drying ratio of 0.45.

## 1 Introduction

Patagonia, here referred to as the area in South America south of the Seno Reloncaví in Chile and Rio Colorado in Argentina, hosts the largest ice fields in the southern hemisphere outside of Antarctica (Meier et al. 2018). Climatic conditions in the area are predominantly influenced by the strong and persistent Southern Hemisphere Westerlies (SHW) (Garreaud et al., 2013). The westerlies take up moisture over the Pacific which precipitates due to pseudoadiabatic cooling when air masses ascend across the Patagonian Andes. In contrast, air masses descend downwind of the Andean main crest resulting in a large hydrographic gradient from west to east in Patagonia that is also evident in the isotopic composition of precipitation (Stern





and Blisniuk, 2002; Smith and Evans, 2007). Besides these orographic effects, different origins of air masses and trajectories have an influence on the isotopic composition of rainfall (Mayr et al., 2007; Grießinger et al., 2018). Backward trajectory calculations identified major moisture sources for the Southern Patagonian Icefield in the western Pacific Ocean (Grießinger et al., 2018). So far no comprehensive study has yet been carried out, which addresses the interplay between isotopic effects

and moisture origin over the total area of Patagonia. Such information is of interest for understanding the regional hydrological cycle and the significance of isotopic proxies used in regional palaeoclimate studies (Moy et al., 2008; Mayr et al., 2013; Zhu et al., 2014). However, the database for the calibration of isotope proxies from this remote region is generally poor. To overcome this lack of data, we have analyzed the hydrogen and oxygen stable isotope values of surface water (lakes and streams) and precipitation collected in Chile and Argentina between 37° S and 55° S and used the available databases for

isotopes in precipitation. In addition, we used reanalysis data for backward trajectory modelling and interpretation of hydro-isotope patterns.

Our particular interest is on isotopic fractionation effects resulting from strong longitudinal and latitudinal hydrological gradients. On the one hand, the Andes form an orographic obstacle perpendicular to the main atmospheric flow resulting in strong longitudinal gradients, e.g. in precipitation amount. On the other hand, different atmospheric flow patterns promote a

latitudinal gradient especially in the transition zone between the SHW and the South American monsoon system (Zhou and Lau, 2001) that influence precipitation in the southern Pampas region bordering Patagonia in the northeast. In particular, the following research questions were addressed:

(1)  Which are the main moisture sources and can they be distinguished isotopically?

(2)  Which imprint do orographic rainout effects have on surface water isotopic composition on the lee side of the Patagonian

Andes?

(3)  How does the regional climate influence surface-water evaporation rates and associated fractionation of hydrogen and oxygen isotopes?

## 2 Material and methods

### 2.1 GNIP data

Monthly $\delta^{18}O$ and $\delta^{2}H$ values of the Global Network of Isotopes in Precipitation (GNIP) of selected precipitation-collection locations were accessed via the International Atomic Energy Agency database (IAEA/WMO, 2018). Isotope data were originally provided to the GNIP database by Instituto de Geocronología y Geología Isotópica, INGEIS (sites Bahia Blanca, Bariloche, Puerto Madryn, Ushuaia) and Comisión Chilena de Energía Nuclear, CCHEN Laboratorio Isótopos ambientales (site Puerto Montt). For reasons of better inter-site comparison only data since 1982 were evaluated, omitting an older period

(1964-1979) which is only available for Puerto Montt. For the calculation of mean isotope values on a yearly basis only years with ≥6 months of data were considered. For those years, for which data of all months were available, averages were weighted with the monthly precipitation amount.



## 2.2 Water sampling

Water samples were collected in an area between 36.9° S to 54.8° S and 62.2° W to 75.4° W (Fig. 1) between 2013 and 2018. The samples included 123 sites of lentic waters (puddles, ponds, lakes), 117 lotic water sites (brooks, rivers), 4 groundwater sites (springs, wells), and 44 single precipitation events. As some sites were sampled repeatedly, the total amount of samples sums up to 339 (Supplementary Table 1). Most of the sample locations are situated downwind on the lee side of the Andes. A few are lentic and precipitation samples were taken upwind, west of the Andes including samples from the Madre de Dios Archipelago, westernmost Patagonia.

## 2.3 Stable isotope analysis

Stable oxygen and hydrogen isotope analyses of water samples were performed by cavity ring-down spectroscopy (L2130-I, Picarro Inc., Santa Clara, CA, USA). About 0.8 µl of sample water was injected into the vaporiser, converted to vapour and transported into the cavity with synthetic air as carrier gas. Water samples were measured in replicate together with internal laboratory standards calibrated against international isotopic reference materials, namely VSMOW, SLAP and GISP (Brand et al., 2014). The isotopic compositions are expressed as δ-values in per mil (‰) as follows in eq. 1:

$$\delta = \left( R_{sample}/R_{standard} - 1 \right) \cdot 1000 \tag{1}$$

with $R_{sample}$ and $R_{standard}$ as isotope ratios ($^{18}O/^{16}O$, $^{2}H/^{1}H$) of sample and standard, respectively. All isotope values of oxygen and hydrogen are reported normalized to the Vienna Standard Mean Ocean Water (VSMOW) - Standard Light Antarctic Precipitation (SLAP) scale. Analytical precision as determined from internal standards was better than ± 0.05 ‰ for $\delta^{18}O$ and ± 0.1 ‰ for $\delta^{2}H$.

## 2.4 Calculation of drying ratios

Based on available isotope data, the atmospheric drying ratio (DR) was calculated. The DR is defined as the ratio of the precipitation amount falling across a mountain range (P) to the initial amount of water vapour upwind ($F_0$) of a mountain range (Smith et al., 2003):

$$DR = P/F_0 \tag{2}$$

The drying ratio can be estimated using the isotope ratios ($^{2}H/^{1}H$ or $^{18}O/^{16}O$) of a site close to the vapour source ($R_{P0}$) and a site downwind of the mountain range ($R_P$):

$$DR = 1 - (R_P/R_{P0})^{1/(\alpha-1)} \tag{3}$$

where α in the exponent signifies the fractionation factor for the phase transition from the vapour to the liquid water phase (Smith et al., 2005).

The same equation in δ-notation reads:

$$DR = 1 - \left( \frac{1000+\delta_P}{1000+\delta_{P0}} \right)^{\frac{1}{(\alpha-1)}} \qquad \text{(Kerr et al. 2014) (4).}$$





## 2.5 Moisture source modelling and climate data

The trajectory calculations are realized by the Lagrangian analysis tool (LAGRANTO) (Wernli and Davies, 1997) using the reanalysis product of the European Centre for Medium-Range Weather Forecasts (ERA-Interim) (Berrisford et al., 2011; Dee et al., 2011; Persson, 2015). The trajectories have been integrated backwards for 18 days starting at 11 equidistant pressure levels from the surface to 500 hPa above ground level of the closest grid point of the respective location. This results in 11 backward trajectory calculations of 18 days starting every ERA-Interim time interval of 6 h over the time period of a selected year. Based on these trajectories, the moisture sources were identified using the technique of Sodemann et al. (2008). ERA Interim data for the period AD 1979-2017 were also used to generate isohyetes and wind vectors over southern South America. Wind vectors 10 m above surface and accumulated precipitation between 30° S and 60° S were also obtained from ERA-Interim and averaged for the period 1979 to 2017. Walter-Lieth climate diagrams (Walter and Lieth, 1967) were created on the basis of the CRU TS3.23 dataset (Harris and Jones, 2015).

## 3 Results

### 3.1 Synoptic constellation in Patagonia

ERA-Interim data show the direction and magnitude of wind vectors in the study area (Fig. 2). Mean wind velocities of up to 10 m s$^{-1}$ occur occur around 48° S to 50° S. The mean flow of the SHW is almost perpendicular to the Andean cordillera and the mountain ridge blocks the moist air masses and leads to intense precipitation along the Chilean precordillera. Mean annual precipitation reaches 4200 mm upwind of the Andes in the Chilean fjord area. In contrast, mean annual total precipitation is below 600 mm downwind in the Argentinean steppe (Fig. 2c). During austral winter months (JJA) the humid zone extends upwind of the Andes to a latitude of about 35° S (Fig. 2a), while in austral summer (DJF) the humid band hardly exceeds 40° S (Fig. 2b). Northward of around 35° S the area that receives low mean annual rainfall (< 600 mm) gradually moves towards the north-east (Fig. 2c), where the so-called South American Arid Diagonal (Bruniard, 1982) crosses the Andes. The increasing influence of south-easterly flow leads to enhanced summer rainfall east of the Andes further north of this latitude (Fig. 2b). Different climatic settings at the southern tip of South America are exemplified by climate diagrams of the five stations selected as representatives from the data of the GNIP network (Fig. 2d-h).

### 3.2 Isotopic composition of precipitation

The five selected GNIP sites are qualified by the number of available data and by representing different climatic settings in Patagonia and at its boundaries. Two other Patagonian sites are not considered here, Punta Arenas and Coyhaique. The record of Punta Arenas contains data judged unreliable by IAEA/WMO (Punta Arenas) and therefore was not used. The GNIP data of Coyhaique is not of relevance here, as the site cannot be classified clearly to a downwind or upwind climatic setting. The station records are located upwind and downwind of the Andes, respectively, over the entire SHW latitudinal range in South America (Figs. 1, 2). Puerto Montt (41.47° S, 72.93° W), Bariloche (41.15° S, 71.33° W), and Puerto Madryn (42.48° S, 65.05°





W) represent Patagonian sites at approximately the same latitude at the northern edge of the SHW zone. These stations are located upwind close to the Pacific, downwind on the leeside of the Andes, and at the Atlantic coast, respectively (Fig. 1). Among them Puerto Montt and Puerto Madryn show comparatively high average $\delta^{18}$O (–6.0 ‰ and –7.2 ‰, respectively) and $\delta^2$H values (–40 ‰ and –57 ‰, respectively), while Bariloche exhibits low values ($\delta^{18}$O: –11.2 ‰; $\delta^2$H: –82 ‰) (Fig. 3).

The IAEA site Bahia Blanca is located in the southern Pampas (38.47° S, 62.16° W) next to the Atlantic coast and exhibits high $\delta^{18}$O (–5.9 ‰) and $\delta^2$H (–39 ‰) values, very similar to Puerto Madryn. In contrast, the southernmost IAEA site Ushuaia on the island of Tierra del Fuego (54.78° S, 68.28° W) has values close to those of Bariloche, the average $\delta^{18}$O and $\delta^2$H of Ushuaia are –11.0 ‰ and –84 ‰, respectively (Fig. 3).

A few years of the GNIP dataset comprise all months. Annual means weighted with monthly precipitation amount were

calculated for those years (open circles in Fig. 3). No marked differences between weighted and unweighted values were observed for stations Ushuaia and Bahia Blanca, while weighted values were slightly lower than unweighted at Puerto Montt, Puerto Madryn, and Bariloche possibly due to a larger seasonality of precipitation at the latter sites compared to the former ones. The relatively restricted dataset, however, does not allow further inferences.

Strikingly, the easternmost site Puerto Madryn shows a similar isotopic composition as Puerto Montt and Bahia Blanca located

at about the same latitude. As might be expected, apart from an "orographic rainout effect", other factors, like varying moisture sources or moisture recycling, must have an influence on the isotopic composition of precipitation.

### 3.3 Moisture sources of precipitation

Backward trajectories were calculated for selected years to investigate the moisture sources at the GNIP sites. For Puerto Montt, Bariloche, and Puerto Madryn, which are at approximately the same latitude, the year 1999 was selected as it is the

only year common to these records. For the comparison of the northernmost and southernmost sites, Bahia Blanca and Ushuaia, respectively, the year 1984 was selected. The isotopic composition of both years represent typical long-term values for the respective sites (Fig. 3). The modelling results clearly show that the majority of the moisture at the sites Puerto Montt and Bariloche originates from the south-eastern Pacific between 30° S and 45° S and moisture sources (> 0.01 mm month$^{-1}$) reach as far as 130° W (Fig. 4c, d). Despite of the similar moisture sources, however, the isotopic values of Puerto Montt and

Bariloche differ largely due to their upwind and downwind locations, respectively, relative to the Andes (Fig. 3). The average 800 hPa geopotential height in Fig. 4 characterizes the mean atmospheric circulation pattern. The air flow is approximately parallel to the isohypses with lower values on its right-hand side (in the southern hemisphere). The stronger the gradient of the isohypses, the higher is the resulting mean wind velocity. A strong poleward decreasing pressure gradient between the subtropical Pacific anticyclone centred around 30° S and the band of cyclones south of 60° S surrounding Antarctica is visible

and indicates the SHW. Moisture originating from the subtropics in the vicinity of the anticyclone is transported south-eastwards by the SHW conveying the moisture to the sites Puerto Montt and Bariloche, respectively. Moisture recycling and moisture sources from the Atlantic play a negligible role at both sites, due to the predominant SHW. A slightly different pattern





was observed for Ushuaia. There, the moisture-uptake regions are shifted southwards reaching 60° S, and the moisture sources are more dispersed in the Pacific. A small fraction of moisture at Ushuaia also originates from southwards in the Drake Passage and Atlantic coastal areas adjacent to Tierra del Fuego (Fig. 4e).

In contrast, at Puerto Madryn hardly any moisture originates from the Pacific nor from the Atlantic Ocean, although the site is located at the Atlantic coast (Fig. 4b). Almost all precipitated moisture results from recycled moisture evaporated in an area between 30° S and 45° S downwind of the Andes in the dry Patagonian and Pampean regions.

At the site Bahia Blanca, 4° north of Puerto Madryn, a large fraction of the precipitated moisture already originated from the Atlantic, while still evaporated water from the interior Argentinean areas east of the Andes serve as another main moisture source (Fig. 4a). Bahia Blanca is already at the northernmost limit of the SHW zone (Fig. 2c) which is readily reflected in the modelled moisture sources and by the importance of Atlantic moisture at this site.

### 3.4 Patagonian surface waters

The $\delta^2$H-versus-$\delta^{18}$O plots of precipitation from GNIP sites clearly distinguish the samples upwind of the Andes from those downwind (Fig. 5c). Sites located upwind of the Patagonian Andes or in the Pampas area generally exhibit higher values than downwind sites. Precipitation values of upwind and downwind sites match the Global Meteoric Water Line (GMWL) fairly well. Most evident for lentic waters (Fig. 5a), the downwind sites and Pampas waters frequently deviate from the GMWL while upwind sites are located on the GMWL. Lotic waters show a similar pattern, albeit the deviations from the GMWL are less pronounced (Fig. 5b).

Lentic waters from specific Patagonian areas were selected to calculate local evaporation lines (LEL) (Fig. 6). Lakes on the Madre de Dios Archipelago (50° S, 75° W; upwind) exhibit comparatively little isotopic variability. The LEL slope produced by these lake samples is very close to that of the GMWL (7.6 versus 8.2, respectively). An average rainfall composition for Madre de Dios Archipelago was calculated from 32 individual rainfalls sampled in 2016 and 2018 ($\delta^{18}$O: –4.6 ‰; $\delta^2$H: –30 ‰; Supplementary Table 1). Considering the errors of the LEL and GMWL, the slopes are statistically not distinguishable. Similarly, Andean lakes close to Bariloche show a small variability and a comparatively high LEL slope of 6.6 (Fig. 6). The Encadenadas Lakes (37° S, 62-63° W) represent the northernmost lakes of our dataset. They are already situated in the Pampas region north of Patagonia and outside of the SHW core zone. The LEL of these lakes shows a slope of 6.2 and intersects the GMWL at a $\delta^{18}$O of –6.1 ‰ and $\delta^2$H of –39 ‰. These values agree with the average isotopic composition of the next GNIP station Bahia Blanca ($\delta^{18}$O: –5.9 ‰; $\delta^2$H: –39 ‰). Further south,Extra-Andean lakes at 41° S next to Bariloche and in the area of Los Glaciares National Park at 49° S to 50° S, lake water isotopes from the dry area downwind of the Andes result in a slope of the LEL of 5.1. Other sites located close to Bariloche plot almost on to the GMWL and show a slope of 6.6 (Fig. 6). These sites are located in the Andes and receive much higher precipitation amounts than the extra-Andean lakes next to Bariloche. The intersection of Bariloche's LEL with the GMWL ($\delta^{18}$O: –11.6 ‰; $\delta^2$H: –85 ‰) matches with the respective meteoric isotopic composition of the GNIP station Bariloche ($\delta^{18}$O: –11.2 ‰; $\delta^2$H: –82 ‰). At Los Glaciares, lentic waters





show the largest isotopic range. The intersection between LEL and GMWL is at –15.3 ‰ ($\delta^{18}$O) and –115 ‰ ($\delta^2$H) and thus close to the averaged isotopic composition of rainwater ($\delta^{18}$O: –15.1 ‰; $\delta^2$H: –116 ‰) collected from April 2015 to April 2016 at a meteorological station next to El Chaltén (49.38° S, 72.94° W) (Fig. 6). In summary, the intersections of the LEL provide an estimate of the regional average isotope composition of precipitation east of the Andes, especially for regions in

which the slopes markedly differ from that of the GMWL.

## 4 Discussion

### 4.1 Meteoric waters and atmospheric drying ratio

Average isotope precipitation values of Patagonian GNIP stations plot on or close to the meteoric water line (Fig. 5c). The position of the long-term average of a precipitation station on the GMWL is primarily controlled by the temperature during

condensation of vapour via the temperature dependent equilibrium fractionation during this phase transition (Clark and Fritz, 1999). In his classical study, Dansgaard (1964) determined slopes of 0.69 ‰ °C$^{-1}$ for $\delta^{18}$O and 5.6 ‰ °C$^{-1}$ for $\delta^2$H in the isotope versus temperature relations of stations covering a latitudinal range from polar to tropical regions. Later, Rozanski et al. (1993) refined a coefficient of 0.58 ‰ °C$^{-1}$ for $\delta^{18}$O in regions with mean annual temperatures ranging between 20 °C and 0 °C, such as Patagonia. A difference of 5 ‰ between the $\delta^{18}$O of precipitation of Bahia Blanca and Bariloche is, however,

not explainable by on average 0.5 °C temperature difference between both sites (Fig. 2 g, h). Rather, this isotopic discrepancy is the result of an orographic isotope effect for the Patagonian sites situated downwind of the Andes in comparison to Bahia Blanca, which is already outside of the core zone of the SHW.

The orographic isotope effect was previously expressed as atmospheric drying ratio (DR) for the Patagonian Andes (Smith and Evans, 2007). A prerequisite for the calculation of a DR is the prevalence of a persistent wind direction and moisture

source as demonstrated for central Patagonia (Figs. 2,4).

The orographic effect on isotopes was derived from a Rayleigh-type distillation process according to the formula

$$R/R_0 = F^{(\alpha-1)} \quad (4),$$

where R denotes the instantaneous water-vapour isotope ratio ($^{18}$O/$^{16}$O or $^2$H/$^1$H) after the fraction 1–$F$ has condensed, $R_0$ the initial isotopic ratio of water vapour, $F$ the remaining fraction of the initial amount of vapour, and $\alpha$ the isotope fractionation

factor between liquid and vapour water given that the atmosphere is vapour saturated when precipitation occurs (Fritz and Clark, 1997). As $\alpha$ depends on temperature (Majoube, 1971; Horita and Wesolowski, 1994), equation (4) requires assumptions about condensation temperatures.

Using equation (2), Smith & Evans (2007) calculated a DR of 0.48 using an assumed condensation temperature of –10 °C and maximum and minimum $\delta^2$H values of stream waters between 40.7° S and 46.7° S latitude. The same approach using $\delta^{18}$O

values provided a DR of 0.56. A critical point in their study was "the inaccessibility of the outer Pacific island" having "the first orography encountered by westerly winds and thus may catch the first rain to fall" (Smith and Evans, 2007). Moreover, stream waters, especially on the dry downwind side of the Andes, may be prone to evaporation despite of their critical data





pre-selection and then do not reliably reflect the isotopic composition of precipitation (Fig. 5b). Finally, the DR was determined from samples of a large latitudinal range not including the zone of maximum wind strength around 50° S (Fig. 2c).

We calculated the DR for temperatures between +10 °C and –45 °C in the SHW core zone at 49–50° S using the average isotope values obtained from sampled rainfall on Madre de Dios Archipelago ($\delta^{18}O$ = –4.9 ‰, $\delta^2H$ = –33‰) as the most

westerly possible site and from average precipitation in the Los Glaciares area immediately east of the Andes ($\delta^{18}O$ = –15.3 ‰, $\delta^2H$ = –115 ‰).The latter value was determined from the intersection between LEL and GMWL (Fig. 6) , a common approach in isotope hydrological studies (e. g. Telmer and Veizer, 2000). Although this approach has recently been questioned in the context of evaporating soil waters (Benettin et al., 2018), our comparisons with GNIP data (Fig. 6) as well as previous results from Patagonia (Mayr et al., 2007) confirm the validity of this approach using lentic waters from Patagonia. As all

GNIP data used plot on the GMWL within measuring uncertainties (Fig. 5c), we also refused to use poorly defined local meteoric water lines (LMWL) instead of the GMWL. Following Stern & Blisniuk (2007) we used the liquid-vapour fractionation factor of Majoube (1971) for temperatures >0 °C. Ice-vapour fractionation factors of Clark and Fritz (1999) for oxygen and hydrogen were tested for temperatures <0 °C. Calculated DR values reach from 0.62 (+10 °C) to 0.44 (–40 °C) when using $\delta^{18}O$ values as input (Fig. 7). The respective DR values using $\delta^2H$ were similar (0.60 and 0.33, respectively), albeit

diverging at lower temperatures. The DR values calculated from $\delta^{18}O$ approached those of $\delta^2H$ when the oxygen isotopic fractionation factors of Majoube (1970) were used for liquid-ice isotope fractionation. In that case a DR of 0.37 resulted at –40 °C. While the actual spatially and seasonally averaged temperature during condensation when air masses cross the Andes is a matter of debate, we here use a value of –10 °C for better comparison with Smith & Evans (2007). The DR at that temperature was 0.46 and 0.44 using oxygen and hydrogen isotopes, respectively, and the liquid-ice fractionation values

Majoube (1970) for oxygen isotopes. Our mean DR of 0.45 from the core zone of the SHW is lower than the average value of 0.52 given by Smith and Evans (2007) for northern Patagonia. We speculate that evaporative enrichment of downwind stream waters could have led to an overestimated DR in their study. Alternatively, a higher topography could explain higher DR (Lenaerts et al., 2014).

Similarly, as for the GNIP station Puerto Madryn, moisture recycling could play an important role for rainfall in the whole

semi-arid Patagonian steppe area (Fig. 8), but rare events of precipitation coming from the Atlantic cannot be excluded totally. In this area, the isotopic differences of precipitation stemming from air masses from westerly directions versus the very rare precipitation events from the east could be evaluated (Mayr et al., 2007). Air masses from the east had $\delta^{18}O$ and $\delta^2H$ values of 8.3 ‰ and 56 ‰, respectively. Thus, precipitation in eastern Patagonia caused by moisture recycling would result in isotope signatures similarly enriched in the heavy isotopes as the rare rainfall of Atlantic origin (Fig. 8).

**4.2 Atmospheric conditions controlling surface water evaporation**

Slopes of LEL ($S_{LEL}$; $\delta^2H$ /$\delta^{18}O$) varied between 5.1 and 7.6 in different regions of Patagonia (Fig. 6). Previous investigations in Patagonia provided a slope of 6.2 for lakes downwind of the Andes and close to the Magellan Strait (51–53° S, 73–69° W)





(Mayr et al., 2007). In the present study, the highest values occurred in humid upwind and the lowest in dry downwind areas. The $S_{LEL}$ is strongly determined by relative humidity (h) (Gonfiantini, 1986; Gat, 1995). The low $S_{LEL}$ observed in the Los Glaciares and eastern Bariloche areas are congruent with modelled values for high-latitude, semi-arid environments with h around 0.65, while the slope observed on Madre de Dios Archipelago readily agrees with theoretical $S_{LEL}$ calculations for a

coastal site with oceanic vapour source and h around 0.80 (Gibson et al., 2014; Anderson et al., 2016). Similar differences were observed for LELs coastal British Columbian versus continental Saskatchewan lakes in Canada (Gibson et al., 2014). As in the case of the Madre de Dios lentic waters, British Columbian lakes plotted very close to the GMWL, while the LEL slope of Saskatchewan lakes was close to 5 similar to the Los Glaciares region.

The evaporation-to-inflow ratio (E/I) strongly determines the position of a lentic water body on the LEL. The highest value on

the LEL determines the endpoint under steady-state conditions for E/I approaching unity. Such conditions may only be reached in terminal lakes in dry environments (Gat and Levy, 1978; Mayr et al, 2007). Low E/I values occur in through-flow lakes with residence times too short for substantial heavy isotope enrichment through evaporation.

## 5 Conclusions

Light stable isotopes of atmospheric vapour are selectively enriched when air masses cross the Andes due to preferential

rainout of heavy water isotopologues. This rainout effect can be described by the drying ratio, and accordingly, the DR of 0.45 inferred from our data is among the highest reported worldwide (Smith and Evans, 2007). The orographic isotope effect on the downwind side of the Andes leads to an approximate 80 ‰ and 10 ‰ decrease of the $\delta^2H$ and $\delta^{18}O$, respectively, of downwind compared to upwind precipitation.

In the westernmost upwind area of Madre de Dios Archipelago, lentic waters' isotopic composition almost plots on the GMWL

due to the high relative humidity prevailing there. Lentic waters from Madre de Dios Archipelago show comparably low variance on the GMWL presumably due to the solely maritime moisture source and the super-humid climate. In contrast, downwind lakes of Patagonia and south-western Pampas plot on LELs with a slope of around 5 due to the low relative humidity under semi-arid climate. The large spread of lentic waters on the LEL at these sites expresses highly variable E/I ratios and high evaporation rates. The extent of the spread seems to be positively related to regional moisture deficits.

Our data describe large isotope variability in both meteoric and surface waters in Patagonia due to orographic effects, moisture recycling, and variable relative humidity. Fig. 9 summarizes the main factors influencing the hydrogen and oxygen isotopic composition of precipitation and surface waters in the SHW core zone of Patagonia around 50°S. We conclude that the Pacific is the main moisture source in upwind and Andean Patagonia also dominating the primary isotope imprint of precipitation, while on downwind sites in the interior of Patagonia descending air masses (Foehn effects), rainout of moisture orographically

enriched in light isotopologues and potentially moisture recycling overrides the isotopic signature of the initial moisture source (Pacific). Additionally, strong evaporation leads to large isotopic variability and heavy isotope enrichment of lentic waters readily visible in a $\delta^2H$-$\delta^{18}O$ plot. Moisture recycling definitely plays a major role for downwind sites in the vicinity of the Atlantic seaboard.



In consequence, detailed isotope studies are a prerequisite for calibration and correct interpretation of isotope proxies from soils (Tuthorn et al., 2014), sediments (Zhu et al., 2014), and tree-rings (Lavergne et al., 2017) from Patagonia. In light of our results further primary data on the isotopic composition of precipitation is urgently needed given the scarce isotopic information available and the high variability imprinted onto precipitation in this area.

*Author contributions.* CM designed the study, HW analysed the isotope samples, LL calculated trajectories and climatological maps, TS provided climate diagrams. CM and LL wrote the manuscript with contributions from AL, WM, CL, JM, and TS. WM carried out GIS work. CM, CL, JM, and GF organized sampling campaigns.

10 *Data availability.* All isotope data obtained in this study are available from Supplementary Table 1. GNIP data are available at http://www-naweb.iaea.org/napc/ih/IHS_resources_gnip.html.

*Competing interests.* The authors declare they have no competing interests.

15 *Acknowledgements.* Sampling was possible due to travel grants from Deutsche Forschungsgemeinschaft (DFG, MA 4235/8-1), CONICET (D103), Universidad de Buenos Aires Grants (UBACyT 20020100100999, 20020150100026BA), BMBF (01DN16025), cooperation project BMBF/MINCYT (AL15/03), and FONDECYT (1150843 and 1161699). TS acknowledges funding from the DFG (SA 2339/3-1). This is publication 159 with contribution from Huinay Scientific Field Station. We thank Hugo Corbella, Jussi Griessinger, Rodrigo S. Martín, Josefina Ramón-Mercau, Emilio Panichini, Lilian Reiss, Ana Srur, 20 Pedro Tiberi, and Rodrigo Torres for help with sampling and assistance in the field. We are grateful to Alejandro Caparós (Parques Nacionales Los Glaciares, El Chaltén, Argentina) and Aquiles Miranda (IMOPAC, Guarello, Chile) for logistic support.

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







**Figure 1: Map of southern South America showing selected GNIP stations and locations of lentic and lotic waters sampled between 2013 and 2018. Base map and glaciated area from Meier et al. (2018) derived from SRTM, LP DACC NASA Version 3, lakes from Messanger et al. (2016). Abbreviations of GNIP stations: BAR: Bariloche, BBL: Bahia Blanca, PMA: Puerto Madryn, PMO: Puerto Montt, USH: Ushuaia.**





**Figure 2: Average wind vectors 10 m above surface with reference arrow and accumulated precipitation over the southern tip of South America during austral winter [JJA] (a), summer [DJF] (b), and the entire year (c) based on ERA Interim data for the period AD 1979-2017. Red dots indicate the position of the evaluated GNIP stations (abbreviations as in Fig. 1). The Walter-Lieth climate diagrams show the summary of climate conditions at the five GNIP sites over the period 1984-2014 (d) to (h).**





**Figure 3: Mean δ¹⁸O (a) and δ²H (b) values of precipitation at five Patagonian GNIP stations (abbreviations as in Fig. 1). Filled circles represent average values from years with data of ≥6 months, open circles show precipitation-amount-weighted averages of years with isotope data for all months.**





**Figure 4: Moisture sources of the GNIP stations Bahia Blanca (a), Puerto Madryn (b), Puerto Montt (c), Bariloche (d), and Ushuaia (e). for 1984 and 1999, respectively. The maximum of the evaporative contribution exceeding the colour bar is given above the figure in mm/month. The colours represent the amount of moisture of each 0.75° grid point contributing to the precipitation at the respective GNIP station (red dot). Geopotential height of the 850 hPa indicates the location of the SHW.**





**Figure 5: Scatterplots of δ²H versus δ¹⁸O values for lentic waters (a), lotic waters (b), annual precipitation from GNIP stations (c). Stars represent Patagonian sites, circles Pampas sites (Encadenadas lakes and GNIP station Bahia Blanca, respectively). The black line represents the GMWL (δ²H = (8.17±0.06) * δ¹⁸O +(10.35 ± 0.65); Rozanski et al. 1993).**




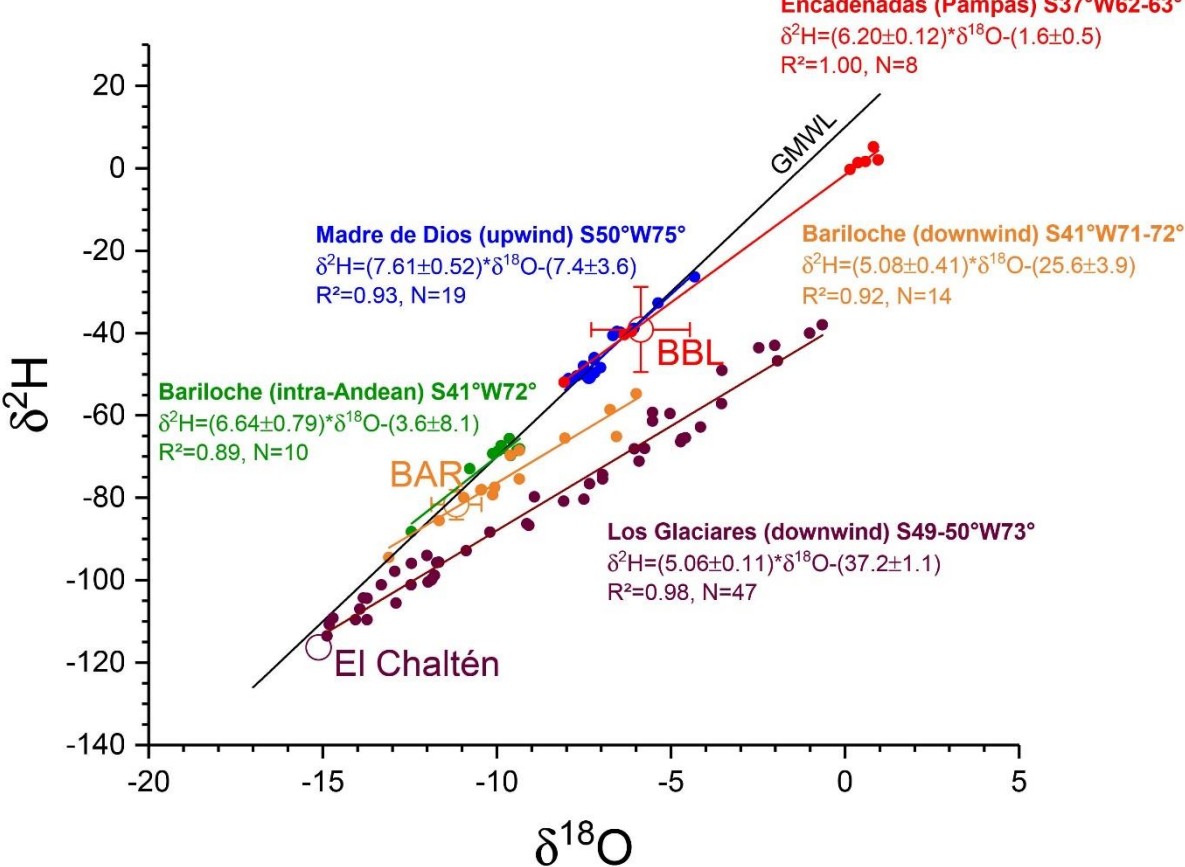

**Figure 6:** $\delta^2H$ versus $\delta^{18}O$ values of lentic waters of selected regions in Patagonia from 37°S to 50°S. Evaporation lines (stippled lines) and respective equations are given. Open symbols represent mean isotopic composition of GNIP stations Bahia Blanca (BBL) and Bariloche (BAR), and of precipitation from El Chaltén and Guarello (bars represent standards deviations of the mean of several years). Note that precipitation values of the three sites coincide with the origin of the evaporation lines on the GMWL.





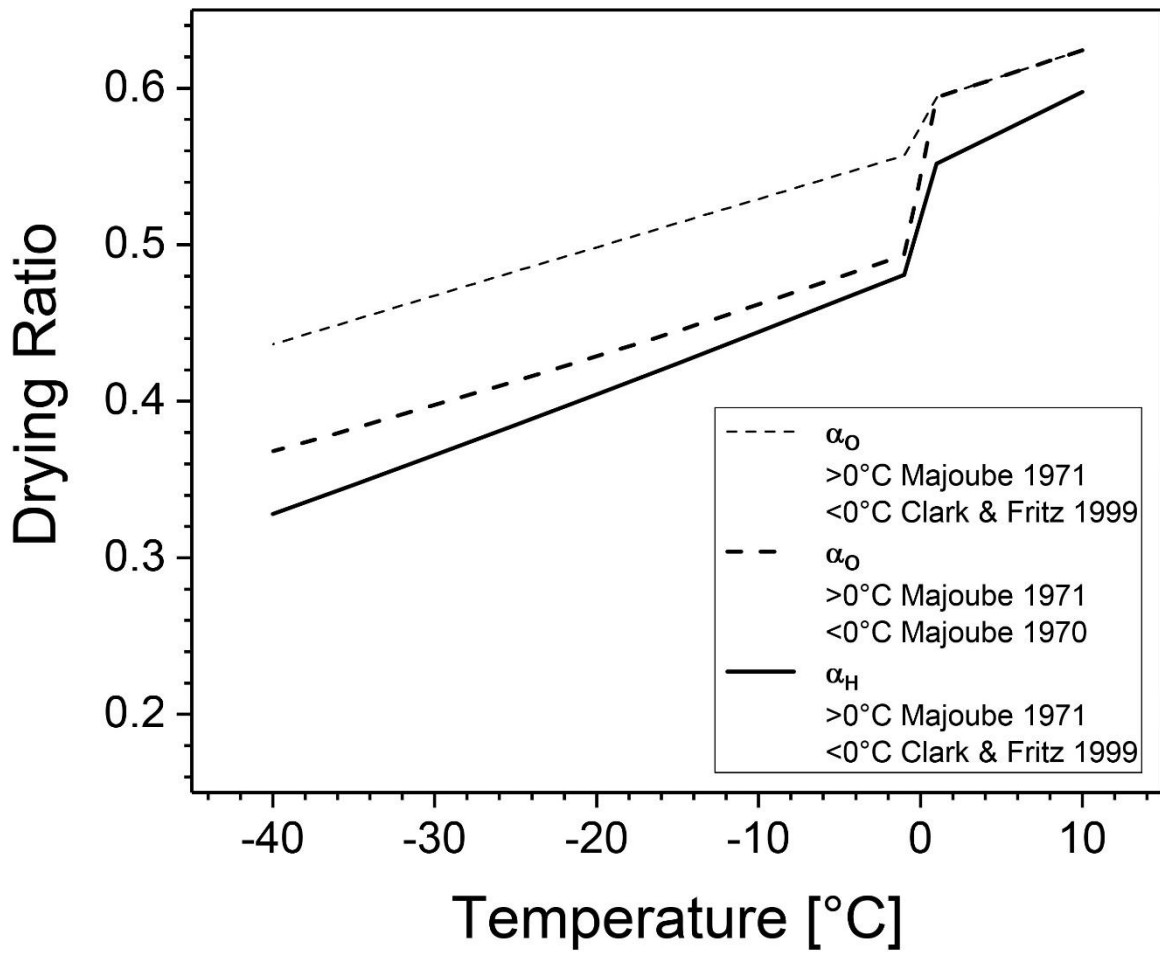

**Figure 7: Drying ratios calculated for varying temperatures using different published fractionation factors of hydrogen ($\alpha_H$) and oxygen ($\alpha_O$).**




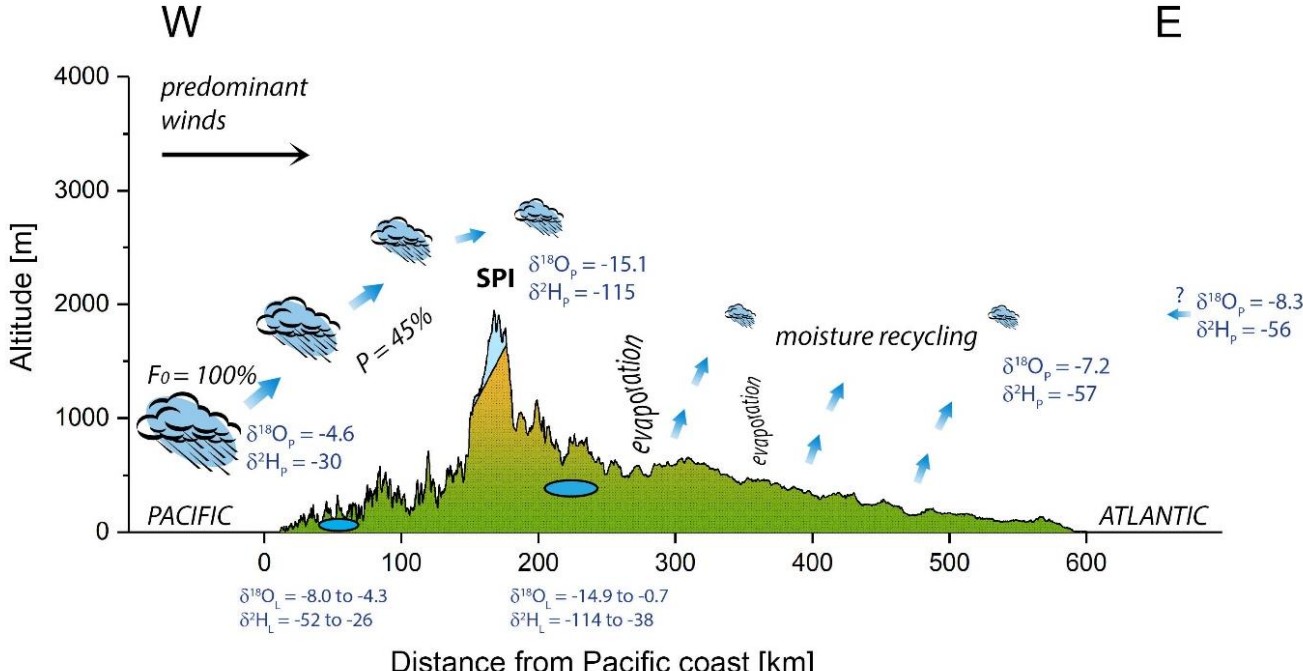

**Figure 8: Schematic section across the South American continent around latitudes 49.0-50.5°S indicating atmospheric influences associated with isotope fractionation and resulting δ values of precipitation ($\delta_P$) and lentic waters ($\delta_L$). Position of the Southern Patagonia Icefield (SPI, mean glacier elevation) and averaged topography is derived from Meier et al. (2018). Precipitation values come from Guarello (mean of 32 individual rainfalls).**




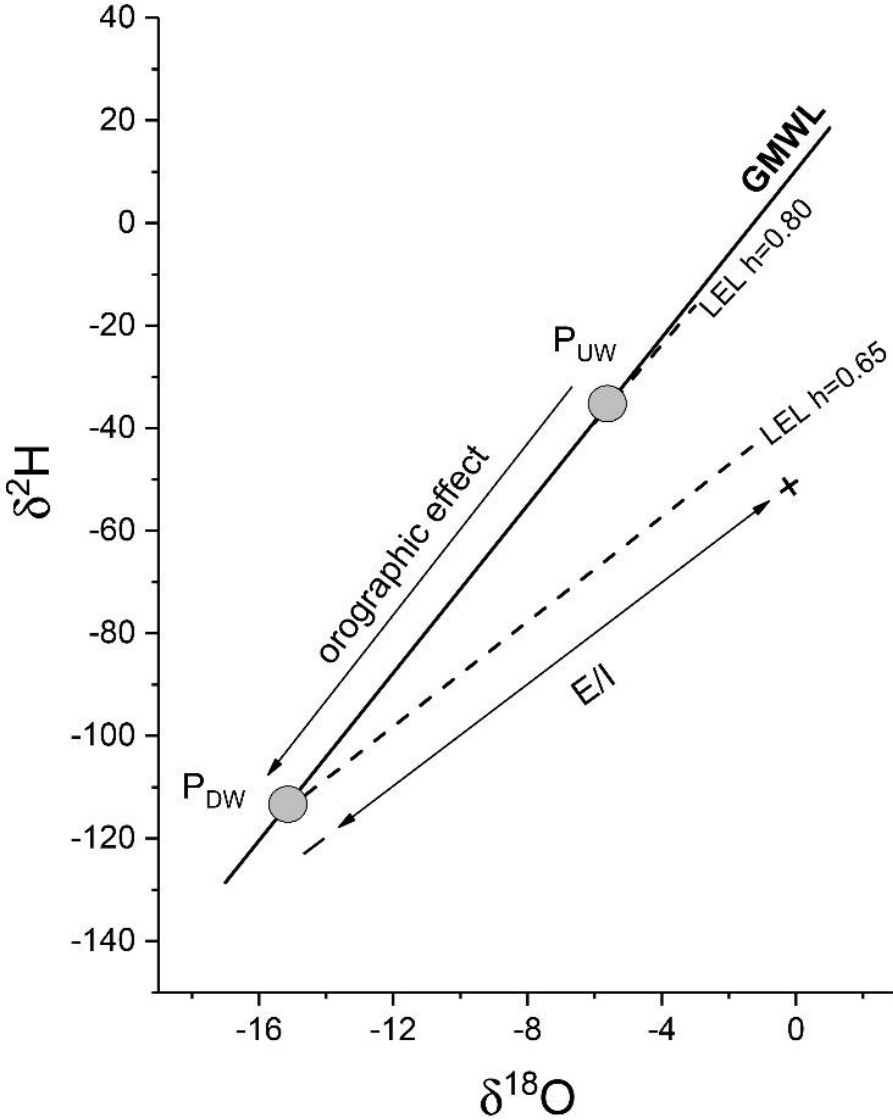

**Figure 9: Summary of isotope effects on precipitation upwind (P_UW) and downwind (P_UW) of the Andes. LEL (stippled lines) originate from these values. Slopes of LEL are determined by the different humidity conditions and their lengths by E/I ratios.**