# Peer review of "Atmospheric controls on hydrogen and oxygen isotope composition of meteoric and surface waters in Patagonia"

_Hydrology and Earth System Sciences, 2018_

## Referee Comment (RC1) · Anonymous Referee #1 · 20 Sep 2018

Title: Atmospheric controls on hydrogen and oxygen isotope composition of meteoric and surface waters in Patagonia

Authors: Christoph Mayr, Lukas Langhamer, Holger Wissel, Wolfgang Meier, Tobias Sauter, Cecilia Laprida, Julieta Massaferro, GuÌĹnter FoÌĹrsterra, and Andreas LuÌĹcke

Recommendation: re-submit

Summary

This paper evaluates the isotopic composition of precipitation, lentic water, and lotic waters to understand the isotopic fractionation processes associated with orographic

rainout, moisture recycling, and moisture sources. The authors found that for sites upwind of the Andes and immediate downwind of Andes, the source of moisture comes from the Pacific Ocean, while for the sites at the Patagonia Atlantic coast regions, the source of moisture comes from the recycled moisture from the continent. The north of Patagonia site obtains moisture from the Atlantic coast. Furthermore, they also relate the rainout effect of water isotopes with the drying ratio and found the decrease of $\delta$D in precipitation around 80 permil from the upwind region to downwind region with drying ratio of 0.45.

Assessment

I have an impression in the beginning that this paper would exploit more on the atmospheric controls on the isotopic composition of surface waters in Patagonia such as rain evaporation, convection, mixing, etc (See my general comments for further explanation). Thus in my opinion the title of the paper is misleading. The works itself are interesting and a lot of efforts have been carried out to collect all the samples. While the work and findings are interesting, some improvements need to be carried out thus the readers will get clear and comprehensive overview on the works. I have several general comments below. I would ask the authors to take these comments into consideration as they re-submit the paper for publication in HESS.

General Comments

1. Based on the title, I thought that the authors would use water vapor isotopic composition in their study to better quantify the atmospheric controls of surface water isotopic composition. In this paper, the authors only analyze the isotopic composition of precipitation (and other surface waters) and analyze moisture sources using backward trajectory model. I also expect to see how the water isotopes evolve from the source (e.g., pacific ocean) to the west side of Patagonia (see Berkelhammer et al., 2012 for example). There is no isotope analysis to confirm the moisture source. It is solely based on moisture backward trajectory analysis.

2. I highly recommend to rewrite the introduction section. I do like short but straight and congested introduction section. However, there are at least two topics needed to be covered in the introduction, thus the readers will have clear insight about the topic. HESS has broad readers and many of them do not have deeply understanding about water isotopes and moisture recycling. I recommend to add at least two paragraphs about: 1. Water isotopes and its fractionation processes, and briefly about Isotopes effects (amount effect, temperature effect, and latitude effect) since it is the main topic in the paper, and 2. Moisture recycling works (e.g., by van der Ent et al., 2010). There are some papers about moisture recycling that are missing in the citation.

3. For the GNIP data, the authors used mean isotope values on a yearly basis if more than 6 months data are available in the particular year. In my opinion it can be tricky since isotope values during summer and winter, and during rainy season and dry season are completely different. If all 6 months isotope data were measured during winter/rainy season, then the $\delta$ values will be lower and do not represent $\delta$ values for that particular year. Do the authors consider this in their analysis?

4. The use of drying ratio to express rainout process is not clear. Since the beginning the authors do not explain what does the drying ratio tell the readers? What is the purpose of using drying ratio in the analysis? I found out later in the conclusion that drying ratio describes the rainout process. Moreover, the authors also do not explain the relation between drying ratio and rainout process. Low drying ratio means low rainout process or is it vice versa? More detailed explanation about drying ratio is needed. In addition, the locations where the authors taken the upwind samples (Madre de Dios and Los Glaciares)) for drying analysis are not stated in Figure 1.

5. I am also puzzled with the use of Rayleigh-type distillation process in the discussion. How the authors used Rayleigh distillation method is unclear. Rayleigh distillation method can be very useful to describe evaporation, rain evaporation, and condensation process during moisture transport (see Worden et al., 2007; Noone, 2012). This is not covered in the study although the formula and few explanations about Rayleigh

method are written in the manuscript.

Line by line comments

L refers to line and P refers to Page.

P1L18-20: Authors stated that they evaluated the stable isotopic compositions of precipitation, lentic waters, and lotic waters to understand isotopic fractionation processes associated with orographic rainout, moisture recycling, and moisture sources. I cannot find in the text how the authors used these water isotopes to understand the processes associated with moisture recycling and moisture sources. They only used backward trajectory to find the moisture sources (see my general comments).

P1L31-32: Rewrite the sentence. Grammatically incorrect.

P3L3: The authors mentioned that they collected 4 groundwater samples. However, I do not see any results on groundwater. So maybe delete this statement?

P3L6: Delete are. Grammatically incorrect.

P3: For calculation of drying ratio, I suggest to write a reason why you use formula number 4 instead number 2 and 3. The reason may be that you do not have water vapor isotopes measurements.

P4L5-7: Rewrite the sentence.

P4L7: You may write:.......using the technique introduced by Sodemann et al. (2008).

P4L27: Maybe remove sentence for two other Patagonian sites that are not considered in the study.

P4L30: Delete respectively.

P5L14-16: For these two sentences, can you elaborate more on the results? For example mention about precipitation amount and temperature (related to altitude). Precipitation amount and temperature strongly influence the isotopic composition of precipitation.

P5L24-25: I am wondering if big difference between $\delta$ precipitation in Puerto Montt (PMO) and Bariloche (BAR) is due to upwind and downwind locations alone. I suggest the authors also relate $\delta$ values with precipitation amount and temperature at that particular year (1999).

P6L18-29: It will help the reader if you write the colors for all the locations. For example: Andean lakes close to Bariloche (Green) show. . ...

P6L29: Here you mention again about Bariloche plot (green). Is it the same with previous sentence (Andean lake)?

P7L8: Rewrite the sentence. Grammatically incorrect.

P7L14-15: Here you mention the $\delta$ difference between Bahia Blanca (BBL) and Bariloche (BAR). But in the next sentence, you mention about temperature difference of 0.5 degree celcius and Figure 2g and 2h. When I look to your Figure 2, then I found BBL is Figure 2g (correct) and BAR is Figure 2e (not correct). The temperature difference is also not correct.

P7L15-17: I will say the discrepancy in $\delta$ is due to temperature different or altitude different, which leads to orographic isotope effect.

P7L21-27: For Rayleigh distillation method, please see my general comments.

P8L4-5: Missing location for Madre de Dios and Los Glaciares. Please see my general comments.

P8L10: Rewrite the sentence. Grammatically incorrect.

P8L19: You may re-write the sentence into:. . .., respectively, using the liquid-ice fractionation values taken from Majoube. . ...

P9L17: Delete word: of downwind.

P9L19: Replace "almost plots on the GMWL" with " are plotted close to GMWL".

P14L2: missing the: the selected GNIP stations.

P14L6: You stated GNIP period from 1984-2014. However, Figure 2 d-f say 1970-2000. Which one is the correct one? Can you also please improve the Figure 2? I can hardly read the wind vectors.

P17L4: The sentence: for 1984 and 1999, respectively is not correct.

P18L4: missing "and". Stars represent Patagonian sites and circles represent Pampas......

P19L2:.....values of lentic waters for selected regions...

P22L2: Change PUW with PDW and change stippled line into dash lines.

References

Berkelhammer, M., Risi, C., Kurita, N., and Noone, D. C.: The moisture source sequence for the Madden-Julian Oscillation as derived from satellite retrievals of HDO and H2O, J. Geophys. Res. Lett., 117, 2012.

Van der Ent, R. J., Savenije, H. H. G., Schaefli, B., and Steele-Dunne, S. C.: Origin and fate of atmospheric moisture over continents, WRR, 46, 2010.

Worden, J., Noone, D., Bowman, K., and TES team.: Importance of rain evaporation and continental convection in the tropical water cycle, Nature, 445, 2007.

Noone, D.: Pairing measurements of the water vapor isotope ratio with humidity to deduce atmospheric moistening and dehydration in the tropical midtroposphere, J. Clim., 25, 2012.

---

## Referee Comment (RC2) · Anonymous Referee #2 · 28 Dec 2018

This paper evaluates the stable isotope compositions of precipitation, lentic waters, and lotic waters in that area to characterize and understand isotope fractionation processes associated with orographic rainout, moisture recycling and moisture sources. I think this paper is relevant since it studies isotopes and the relevant literature cited, however it requires major revisions. The manuscript is interesting but written is poor. It is difficult to catch the authors' messages to readers. The results are original and represent an important contribution to the understanding of the controls on hydrogen and oxygen isotope composition. However, my main concern is that the problem statement is not clearly defined and that the field description is not sufficient as it is. Specific comments: Abstract: This is mostly composed of the Introduction and Material and methods. So,

[Figure]

I do not think the abstract is attractive. Introduction: Describe more paper about the stable isotope compositions! which problems? why is this study important? Material and methods: Detail more the monitoring and describe the collection methodology. I think the paper would be better if the authors combined the results and discussion.

---

## Author Comment (AC1) · 4 Jan 2019

Dear editors and reviewers of manuscript HESS-2018-431

we gratefully acknowledge the two reviews of our manuscript and would like to respond to the points of critique in detail before the closing date for open discussion. By doing so, we hope we can clarify some topics and stimulate discussion. We listed below all comments of the two reviewers and our detailed response below.

Kind regards on behalf of all authors, Christoph Mayr

Reply to comments of Reviewer 1:
Recommendation: re-submit Summary This paper evaluates the isotopic composition of precipitation, lentic water, and lotic waters to understand the isotopic fractionation processes associated with orographic rainout, moisture recycling, and moisture sources. The authors found that for sites upwind of the Andes and immediate downwind of Andes, the source of moisture comes from the Pacific Ocean, while for the sites at the Patagonia Atlantic coast regions, the source of moisture comes from the recycled moisture from the continent. The north of Patagonia site obtains moisture from the Atlantic coast. Furthermore, they also relate the rainout effect of water isotopes with the drying ratio and found the decrease of $\delta$D in precipitation around 80 permil from the upwind region to downwind region with drying ratio of 0.45. Assessment I have an impression in the beginning that this paper would exploit more on the atmospheric controls on the isotopic composition of surface waters in Patagonia such as rain evaporation, convection, mixing, etc (See my general comments for further explanation). Thus in my opinion the title of the paper is misleading.

Reply: Although we think that the main drivers for the observed isotope variations, i.e. rainout, moisture sources and humidity changes, are indeed related to atmospheric controls, we agree that a more general title is also adequate. Therefore, we suggest to rephrase the title: "Controls on hydrogen and oxygen isotope composition of meteoric and surface waters in Patagonia".

The works itself are interesting and a lot of efforts have been carried out to collect all the samples. While the work and findings are interesting, some improvements need to be carried out thus the readers will get clear and comprehensive overview on the works. I have several general comments below. I would ask the authors to take these comments into consideration as they re-submit the paper for publication in HESS. General Comments 1. Based on the title, I thought that the authors would use water vapor isotopic composition in their study to better quantify the atmospheric controls of surface water isotopic composition. In this paper, the authors only analyze the isotopic composition of precipitation (and other surface waters) and analyze moisture sources

using backward trajectory model. I also expect to see how the water isotopes evolve from the source (e.g., pacific ocean) to the west side of Patagonia (see Berkelhammer et al., 2012 for example). There is no isotope analysis to confirm the moisture source. It is solely based on moisture backward trajectory analysis.

Reply: The reviewer doubts that we were able to quantify atmospheric controls. However, the primary purpose of our study is a qualitative description of the fractionation processes as schematically shown in Fig. 9, and not a quantification. Nevertheless, quantification is partly possible as shown by the humidity estimations that determine the slopes of the evaporation lines. We consider near-ground humidity as an atmospheric control and therefore we do not agree that the manuscript does not deal with "atmospheric controls". However, the orographic effect may or may not be considered an atmospheric control and therefore we changed the title as already stated above. The reviewer states that we "only" analysed the isotopic composition of precipitation. Actually, the precipitation data are only a small fraction of the results we present in the manuscript. We contrast those data with a huge data set of isotopic composition of surface waters to infer the isotopic composition of precipitation at remote places where this data (direct measurements) is unavailable. These places, however, are the most interesting to infer the maximum drying ratio of the Andes, because they are located in the core of the Southern Westerlies. We additionally use the GNIP data to verify our assumptions about moisture sources. The intention of this approach is not primarily to use isotopes for moisture trajectory reconstructions, but rather to demonstrate that different moisture source areas and atmospheric pathways could create similar isotopic compositions in precipitation as is the case for PMO, BB and PMA in our study. Apparently this was not stated clearly enough in the text and therefore we will add a respective paragraph. Berkelhammer et al. (2012) used a fundamentally different approach by evaluating satellite data. In their study, there is neither ground-truth data given that the tropospheric ïĄ∂2H variations of vapour do reliably reflect measured ïĄ∂2H values of precipitation near ground, nor is it proven that isotope variability on the scale of the Andes orogen are reliably captured by this method. Moreover, the

"traditional" way of analysing isotopes in surface waters, that we use in our study, integrates over longer time scales (i.e., residence times of lakes) than the given reference. The main information in our study comes from the combination of ïĄď18O AND ïĄď2H values as illustrated in our Fig. 9. Therefore, not only methodologies, but also spatial and temporal scales are different in our study than in Berkelhammer et al. (2012). Both methodologies may complement each other, but finding this out is beyond the focus of our study and remains to be explored in future approaches.

2. I highly recommend to rewrite the introduction section. I do like short but straight and congested introduction section. However, there are at least two topics needed to be covered in the introduction, thus the readers will have clear insight about the topic. HESS has broad readers and many of them do not have deeply understanding about water isotopes and moisture recycling. I recommend to add at least two paragraphs about: 1. Water isotopes and its fractionation processes, and briefly about Isotopes effects (amount effect, temperature effect, and latitude effect) since it is the main topic in the paper, and 2. Moisture recycling works (e.g., by van der Ent et al., 2010). There are some papers about moisture recycling that are missing in the citation.

Reply: We agree that it is useful to add the requested information and will do so in a revised version.

3. For the GNIP data, the authors used mean isotope values on a yearly basis if more than 6 months data are available in the particular year. In my opinion it can be tricky since isotope values during summer and winter, and during rainy season and dry season are completely different. If all 6 months isotope data were measured during winter/rainy season, then the $\delta$ values will be lower and do not represent $\delta$ values for that particular year. Do the authors consider this in their analysis?

Reply: Actually, we did consider it by adding the annual amount-weighted isotope value for years for which all 12 months were available. As stated in the text, there are only few such GNIP data points for which this information is available, but for those a comparison

with the values of less than 12 months has been made in Fig. 3 (open compared to closed symbols). We could additionally provide a figure with average monthly isotope variations over the year to illustrate seasonal differences. However, the drying ratio at 50°S has been calculated without using GNIP data, as we used the surface waters to infer precipitation isotope values instead. Therefore, our main result, i.e. the calculation of the drying ratio at 50°S, is anyway independent from the GNIP data.

4. The use of drying ratio to express rainout process is not clear. Since the beginning the authors do not explain what does the drying ratio tell the readers? What is the purpose of using drying ratio in the analysis? I found out later in the conclusion that drying ratio describes the rainout process. Moreover, the authors also do not explain the relation between drying ratio and rainout process. Low drying ratio means low rainout process or is it vice versa? More detailed explanation about drying ratio is needed. In addition, the locations where the authors taken the upwind samples (Madre de Dios and Los Glaciares) for drying analysis are not stated in Figure 1.

Reply: We totally agree that the locations of Madre de Dios and Los Glaciares should be added in Figure 1. Actually the inserts show the Los Glaciares sites, but this is not clear to readers. We will also add the requested information about the significance of the drying ratio in the introduction or discussion.

5. I am also puzzled with the use of Rayleigh-type distillation process in the discussion. How the authors used Rayleigh distillation method is unclear. Rayleigh distillation method can be very useful to describe evaporation, rain evaporation, and condensation process during moisture transport (see Worden et al., 2007; Noone, 2012). This is not covered in the study although the formula and few explanations about Rayleigh method are written in the manuscript.

Reply: Rayleigh-type models in isotope hydrology generally describe the isotope fractionation that occurs during evaporation in a more or less hydrologically open system, such as evaporation from a lake surface. The orographic effect also can be seen as

such an open system, as hardly any of the precipitating moisture re-enters the transported atmospheric moisture and kinetic fractionation dominates this process. In contrast, equilibrium fractionation is the dominant process for isotope separation during rain formation. We will add more explaining text in a revised version and adequate references for better understanding of the underlying principles.

Line by line comments L refers to line and P refers to Page. P1L18-20: Authors stated that they evaluated the stable isotopic compositions of precipitation, lentic waters, and lotic waters to understand isotopic fractionation processes associated with orographic rainout, moisture recycling, and moisture sources. I cannot find in the text how the authors used these water isotopes to understand the processes associated with moisture recycling and moisture sources. They only used backward trajectory to find the moisture sources (see my general comments).

Reply: We do not agree with the reviewer that we have not used our data set to discuss processes of moisture recycling and moisture sources (see also our reply to the general comment). Indeed, the processes, and how we understand them, are summarized in Fig. 9. These processes are related to source water variations (due to different trajectories, orographic effects), temperature (influencing isotope values of upwind precipitation via equilibrium fractionation), and evaporation (influencing lake water isotope composition). Obviously our intention and the interplay of these processes are not clearly formulated for a non-specialized audience and we will clarify this in a revised version.

P1L31-32: Rewrite the sentence. Grammatically incorrect.

Reply: Will be done in the revised version.

P3L3: The authors mentioned that they collected 4 groundwater samples. However, I do not see any results on groundwater. So maybe delete this statement?

Reply: Will be deleted in the revised version.

P3L6: Delete are. Grammatically incorrect.

Reply: Will be rephrased in the revised version.

P3: For calculation of drying ratio, I suggest to write a reason why you use formula number 4 instead number 2 and 3. The reason may be that you do not have water vapor isotopes measurements.

Reply: Actually, both equations 3 and 4 can be used. We just want to demonstrate that different formulas are published that essentially give the same results. This will be more clearly expressed in a revised version.

P4L5-7: Rewrite the sentence.

Reply: Will be rephrased in the revised version.

P4L7: You may write: .......using the technique introduced by Sodemann et al. (2008).

Reply: Will be rephrased according to the reviewer's suggestion in the revised version.

P4L27: Maybe remove sentence for two other Patagonian sites that are not considered in the study.

Reply: Sentence will be removed according to the reviewer's suggestion in the revised version.

P4L30: Delete respectively.

Reply: Will be deleted in the revised version.

P5L14-16: For these two sentences, can you elaborate more on the results? For example, mention about precipitation amount and temperature (related to altitude). Precipitation amount and temperature strongly influence the isotopic composition of precipitation.

Reply: We can add this information in the revised version.

P5L24-25: I am wondering if big difference between $\delta$ precipitation in Puerto Montt (PMO) and Bariloche (BAR) is due to upwind and downwind locations alone. I suggest the authors also relate $\delta$ values with precipitation amount and temperature at that particular year (1999).

Reply: We will add this information in the revised version.

P6L18-29: It will help the reader if you write the colors for all the locations. For example: Andean lakes close to Bariloche (Green) show ....

Reply: Colours will be added in the revised version.

P6L29: Here you mention again about Bariloche plot (green). Is it the same with previous sentence (Andean lake)?

Reply: We suggest showing the exact positions of the Andean and Extra-Andean lakes e.g. in the inserts of Fig. 1.

P7L8: Rewrite the sentence. Grammatically incorrect.

Reply: Will be rephrased in the revised version.

P7L14-15: Here you mention the $\delta$ difference between Bahia Blanca (BBL) and Bariloche (BAR). But in the next sentence, you mention about temperature difference of 0.5 degree celcius and Figure 2g and 2h. When I look to your Figure 2, then I found BBL is Figure 2g (correct) and BAR is Figure 2e (not correct). The temperature difference is also not correct.

Reply: The reviewer is right that there is a mistake in the submitted text. The comparison refers to Bahia Blanca and Puerto Madryn (not Bariloche). The reference to Fig 2g and 2h therefore is correct.

P7L15-17: I will say the discrepancy in $\delta$ is due to temperature different or altitude different, which leads to orographic isotope effect.

Reply: The difference between Bahia Blanca and Puerto Madryn is not explainable by temperature or altitudinal effects as both are at approximately the same altitude and have similar temperatures. Therefore, the statements are correct except that we erroneously mentioned Bariloche instead of Puerto Madryn. We regret that this led to confusion.

P7L21-27: For Rayleigh distillation method, please see my general comments.

Reply: We will add additional explanations about Rayleigh isotope factionation in a revised version.

P8L4-5: Missing location for Madre de Dios and Los Glaciares. Please see my general comments.

Reply: We accept this point of critique and will add the sites in the map (Fig. 1).

P8L10: Rewrite the sentence. Grammatically incorrect.

Reply: We will rephrase the sentence in the revised version.

P8L19: You may re-write the sentence into: ...., respectively, using the liquid-ice frac-tionation values taken from Majoube.....

Reply: We will rephrase the sentence in the revised version according to the reviewer's suggestions.

P9L17: Delete word: of downwind.

Reply: We will delete it in the revised version.

P9L19: Replace "almost plots on the GMWL" with " are plotted close to GMWL".

Reply: We will rephrase the sentence in the revised version.

P14L2: missing the: the selected GNIP stations.

Reply: We will rephrase it in the revised version.

P14L6: You stated GNIP period from 1984-2014. However, Figure 2 d-f say 1970-2000. Which one is the correct one? Can you also please improve the Figure 2? I can hardly read the wind vectors.

Reply: Fig. 2d-f is not from the GNIP dataset; therefore, the time interval is different (1970-2000) and the time interval mentioned in the caption is indeed wrong. In the revised version we will adapt the time interval. We will also generally improve the figure.

P17L4: The sentence: for 1984 and 1999, respectively is not correct.

Reply: We will remove the full stop before "for 1984 and 1999, respectively."

P18L4: missing "and". Stars represent Patagonian sites and circles represent Pampas ......

Reply: We will add the "and".

P19L2: ....values of lentic waters for selected regions...

Reply: We will replace "of" with "for".

P22L2: Change PUW with PDW and change stippled line into dash lines.

Reply: We will change the second "PUW" in the figure caption with "PDW" and "stippled" with "dashed".

References Berkelhammer, M., Risi, C., Kurita, N., and Noone, D. C.: The moisture source sequence for the Madden-Julian Oscillation as derived from satellite retrievals of HDO and H2O, J. Geophys. Res. Lett., 117, 2012. Van der Ent, R. J., Savenije, H. H. G., Schaefli, B., and Steele-Dunne, S. C.: Origin and fate of atmospheric moisture over continents, WRR, 46, 2010. Worden, J., Noone, D., Bowman, K., and TES team.: Importance of rain evaporation and continental convection in the tropical water cycle, Nature, 445, 2007. Noone, D.: Pairing measurements of the water vapor isotope ratio with humidity to deduce atmospheric moistening and dehydration in the tropical

midtroposphere, J. Clim., 25, 2012.

Reply: For the sake of completeness we will add some of the listed references where appropriate. However, we need to stress here again that the approaches and results of these references are not directly comparable to the results of our study. As a direct comparison is not possible and beyond the scope of our study, we do not intend to exhaustively discuss the satellite-inferred data.

Reply to comments of Reviewer 2:

This paper evaluates the stable isotope compositions of precipitation, lentic waters, and lotic waters in that area to characterize and understand isotope fractionation processes associated with orographic rainout, moisture recycling and moisture sources. I think this paper is relevant since it studies isotopes and the relevant literature cited, however it requires major revisions. The manuscript is interesting but written is poor. It is difficult to catch the authors' messages to readers.

Reply: The critical comments at the end of the paragraph unfortunately are rather unspecific. Thus, it is not possible to directly respond. There may be linguistic flaws in the text, but no example is given by Reviewer 2. The message to the readers will be improved by adding more background details, as suggested by the reviewer in the specific comments.

The results are original and represent an important contribution to the understanding of the controls on hydrogen and oxygen isotope composition. However, my main concern is that the problem statement is not clearly defined and that the field description is not sufficient as it is.

Reply: We will clarify the information on the research objectives and the field description in a revised version.

Specific comments: Abstract: This is mostly composed of the Introduction and Material and methods. So, I do not think the abstract is attractive.

Reply: We agree that our results and conclusions should be highlighted in more detail in the abstract. We will do so in a revised version.

Introduction: Describe more paper about the stable isotope compositions! which problems? why is this study important?

Reply: We agree that this part could be more elaborated and will add additional information in a revised version.

Material and methods: Detail more the monitoring and describe the collection methodology.

Reply: We can add additional methodological details in a revised version.

I think the paper would be better if the authors combined the results and discussion.

Reply: We do not see a reason in the reviewer's comment for merging results and discussion to one chapter and prefer the chapter structure as it is.

---

## Author Comment (AC2) · 30 Jan 2019

Replies to comments of Reviewer 1:

Reviewer 1: Summary This paper evaluates the isotopic composition of precipitation, lentic water, and lotic waters to understand the isotopic fractionation processes associated with orographic rainout, moisture recycling, and moisture sources. The authors found that for sites upwind of the Andes and immediate downwind of Andes, the source of moisture comes from the Pacific Ocean, while for the sites at the Patagonia Atlantic coast regions, the source of moisture comes from the recycled moisture from the continent. The north of Patagonia site obtains moisture from the Atlantic coast.

Furthermore, they also relate the rainout effect of water isotopes with the drying ratio and found the decrease of $\delta$D in precipitation around 80 permil from the upwind region to downwind region with drying ratio of 0.45. Assessment I have an impression in the beginning that this paper would exploit more on the atmospheric controls on the isotopic composition of surface waters in Patagonia such as rain evaporation, convection, mixing, etc (See my general comments for further explanation). Thus in my opinion the title of the paper is misleading.

Reply: The main drivers for the observed isotope variations, i.e. rainout, moisture sources and humidity changes, are controlled by atmospheric drivers and therefore we chose the title. However, we agree that a more general title is also appropriate. We suggest to modify the title accordingly: "Controls on hydrogen and oxygen isotope composition of meteoric and surface waters in Patagonia".

Reviewer 1: The works itself are interesting and a lot of efforts have been carried out to collect all the samples. While the work and findings are interesting, some improvements need to be carried out thus the readers will get clear and comprehensive overview on the works. I have several general comments below. I would ask the authors to take these comments into consideration as they re-submit the paper for publication in HESS. General Comments 1. Based on the title, I thought that the authors would use water vapor isotopic composition in their study to better quantify the atmospheric controls of surface water isotopic composition. In this paper, the authors only analyze the isotopic composition of precipitation (and other surface waters) and analyze moisture sources using backward trajectory model. I also expect to see how the water isotopes evolve from the source (e.g., pacific ocean) to the west side of Patagonia (see Berkelhammer et al., 2012 for example). There is no isotope analysis to confirm the moisture source. It is solely based on moisture backward trajectory analysis.

Reply: The reviewer doubts that we were able to quantify atmospheric controls. However, the primary purpose of our study is a qualitative description of the fractionation processes as schematically shown in Fig. 9, and not a quantification of water vapour

isotope composition. Nevertheless, a quantification of meteorological parameters is partly possible as demonstrated by the humidity estimations that determine the slopes of the evaporation lines. We consider near-ground humidity as an atmospheric control and therefore we do not agree that the manuscript does not deal with "atmospheric controls". As already stated above we will nevertheless change the title to avoid any eventually misleading inferences from the title. The reviewer states that we "only" analysed the isotopic composition of precipitation. Actually, the precipitation data are only a small fraction of the results we present in the manuscript. We contrast those data with a huge data set of isotopic composition of surface waters to infer the isotopic composition of precipitation at remote places where this data (direct measurements) is unavailable. These places, however, are the most interesting to infer the maximum drying ratio of the Andes, because they are located in the core of the Southern Westerlies. We additionally use the GNIP data to verify our assumptions about moisture sources. The intention of this approach is not primarily to use isotopes for moisture trajectory reconstructions, but rather to demonstrate that different moisture source areas and atmospheric pathways could create similar isotopic compositions in precipitation as is the case for PMO, BB and PMA in our study. Apparently this was not stated clearly enough in the text and therefore we will add a respective paragraph. Berkelhammer et al. (2012) used a fundamentally different approach by evaluating satellite data. In their study, there is neither ground truth given that the tropospheric $\delta 2H$ variations of vapour do reliably reflect measured $\delta 2H$ values of precipitation near ground, nor is it proven that isotope variability on the spatial scale of the Andes orogen are reliably captured by this method. Moreover, the "traditional" way of analysing isotopes in surface waters, that we use in our study, integrates over longer time scales (i.e. residence times of lakes) than the given reference. The main information in our study comes from the combination of $\delta 18O$ AND $\delta 2H$ values as illustrated in our Fig. 9. Therefore, not only methodologies, but also spatial and temporal scales are different in our study than in Berkelhammer et al. (2012). Both methodologies may complement each other, but finding out their complementarities is clearly beyond the focus of our study and remains to be explored in future approaches. Without ground truth for the satellite-based approach, we consider the traditional isotope analyses as still being state-of-the-art.

Reviewer 1: 2. I highly recommend to rewrite the introduction section. I do like short but straight and congested introduction section. However, there are at least two topics needed to be covered in the introduction, thus the readers will have clear insight about the topic. HESS has broad readers and many of them do not have deeply understanding about water isotopes and moisture recycling. I recommend to add at least two paragraphs about: 1. Water isotopes and its fractionation processes, and briefly about Isotopes effects (amount effect, temperature effect, and latitude effect) since it is the main topic in the paper, and 2. Moisture recycling works (e.g., by van der Ent et al., 2010). There are some papers about moisture recycling that are missing in the citation.

Reply: We agree that it is useful to add the requested information and will do so in a revised version.

Reviewer 1: 3. For the GNIP data, the authors used mean isotope values on a yearly basis if more than 6 months data are available in the particular year. In my opinion it can be tricky since isotope values during summer and winter, and during rainy season and dry season are completely different. If all 6 months isotope data were measured during winter/rainy season, then the $\delta$ values will be lower and do not represent $\delta$ values for that particular year. Do the authors consider this in their analysis?

Reply: Actually, we did consider it by presenting also the annual amount-weighted isotope value for years for which all 12 months were available in Fig. 3 (open circles). As stated in the text, there are only few such GNIP data points for which this information is available, but for those a comparison with the values of less than 12 months has been made in Fig. 3 (open compared to closed symbols). We could additionally provide a figure with average monthly isotope variations over the year to illustrate seasonal differences. However, the drying ratio at 50°S has been calculated without using GNIP data,

as we used the surface waters to infer precipitation isotope values instead. Therefore, our main result, i.e. the calculation of the drying ratio at 50°S, is independent from the GNIP data anyway.

Reviewer 1: 4. The use of drying ratio to express rainout process is not clear. Since the beginning the authors do not explain what does the drying ratio tell the readers? What is the purpose of using drying ratio in the analysis? I found out later in the conclusion that drying ratio describes the rainout process. Moreover, the authors also do not explain the relation between drying ratio and rainout process. Low drying ratio means low rainout process or is it vice versa? More detailed explanation about drying ratio is needed. In addition, the locations where the authors taken the upwind samples (Madre de Dios and Los Glaciares) for drying analysis are not stated in Figure 1.

Reply: We will add the requested information about the significance of the drying ratio in the introduction or discussion. We also totally agree that the locations of Madre de Dios and Los Glaciares should be added in Figure 1. Actually the inserts show the Los Glaciares sites, but this is not clear to readers. We will rearrange the graphics to make the respective locations more apparent.

Reviewer 1: 5. I am also puzzled with the use of Rayleigh-type distillation process in the discussion. How the authors used Rayleigh distillation method is unclear. Rayleigh distillation method can be very useful to describe evaporation, rain evaporation, and condensation process during moisture transport (see Worden et al., 2007; Noone, 2012). This is not covered in the study although the formula and few explanations about Rayleigh method are written in the manuscript.

Reply: Rayleigh-type models in isotope hydrology generally describe the isotope fractionation that occurs during evaporation in a more or less hydrologically open system, such as evaporation from a lake surface. The orographic effect also can be seen as such an open system, as hardly any of the precipitating moisture re-enters into the transported atmospheric moisture and kinetic fractionation dominates this process. In

contrast, equilibrium fractionation is the dominant process for isotope separation during rain formation. We will add more explaining text in a revised version and adequate references for better understanding of the underlying principles.

Reviewer 1: Line by line comments L refers to line and P refers to Page. P1L18-20: Authors stated that they evaluated the stable isotopic compositions of precipitation, lentic waters, and lotic waters to understand isotopic fractionation processes associated with orographic rainout, moisture recycling, and moisture sources. I cannot find in the text how the authors used these water isotopes to understand the processes associated with moisture recycling and moisture sources. They only used backward trajectory to find the moisture sources (see my general comments).

Reply: We do not agree with the reviewer that we have not used our data set to discuss processes of moisture recycling and moisture sources (see also our reply to the general comment). Indeed, the processes, and how we understand them, are summarized in Fig. 9. These processes are related to source water variations (due to different trajectories, orographic effects), temperature (influencing isotope values of upwind precipitation via equilibrium fractionation), and evaporation (influencing lake water isotope composition). Obviously our intention and the interplay of these processes are not clearly formulated for a non-specialized audience and we will clarify this in a revised version.

Reviewer 1: P1L31-32: Rewrite the sentence. Grammatically incorrect.

Reply: We will rephrase: "Downwind of the Andean main crest, the descending air masses lead to a foehn effect. The result is a large hydrographic gradient from west to east in Patagonia that is also evident in the isotopic composition of precipitation (Stern and Blisniuk, 2002; Smith and Evans, 2007)."

Reviewer 1: P3L3: The authors mentioned that they collected 4 groundwater samples. However, I do not see any results on groundwater. So maybe delete this statement?

Reply: The statement will be deleted in the revised version.

Reviewer 1: P3L6: Delete are. Grammatically incorrect.

Reply: We will rephrase: "Most of the sample locations situated downwind on the lee side of the Andes."

Reviewer 1: P3: For calculation of drying ratio, I suggest to write a reason why you use formula number 4 instead number 2 and 3. The reason may be that you do not have water vapor isotopes measurements.

Reply: Actually, both equations 3 and 4 can be used. We just want to demonstrate that different formulas are published that essentially give the same results. The difference is the use of isotope ratios (R) in the one and $\delta$-values in the other formula, as is clearly stated in the text. As RP/RP0 is the same as $(1000+\delta P)/(1000+\delta P0)$ by definition, both equations actually express the same. The water vapour isotope measurements are not necessary for any of the equations.

Reviewer 1: P4L5-7: Rewrite the sentence.

Reply: We will rephrase the sentence: "Such backward trajectory calculations were calculated for every ERA-Interim time interval of 6 h over the time period of a selected year."

Reviewer 1: P4L7: You may write: .......using the technique introduced by Sodemann et al. (2008).

Reply: We will rephrase it: "Based on these trajectories, the moisture sources were identified using the technique introduced by Sodemann et al. (2008)."

Reviewer 1: P4L27: Maybe remove sentence for two other Patagonian sites that are not considered in the study. P4L30: Delete respectively.

Reply: The sentences regarding the two sites that were not considered will be removed in the revised version according to the reviewer's suggestion.

Reviewer 1: P5L14-16: For these two sentences, can you elaborate more on the results? For example, mention about precipitation amount and temperature (related to altitude). Precipitation amount and temperature strongly influence the isotopic composition of precipitation.

Reply: Precipitation amount and temperature are rather different at the 3 sites. If precipitation and temperature were the main drivers, we would expect very different isotope values. For a better understanding we complemented the sentence: "Strikingly, the easternmost site Puerto Madryn shows a similar isotopic composition as Puerto Montt and Bahia Blanca located at about the same latitude, although temperature and precipitation amount are rather different at these sites (Fig. 2)."

Reviewer 1: P5L24-25: I am wondering if big difference between $\delta$ precipitation in Puerto Montt (PMO) and Bariloche (BAR) is due to upwind and downwind locations alone. I suggest the authors also relate $\delta$ values with precipitation amount and temperature at that particular year (1999).

Reply: We will add this information in the revised version.

Reviewer 1: P6L18-29: It will help the reader if you write the colors for all the locations. For example: Andean lakes close to Bariloche (Green) show ....

Reply: Colours will be added in the revised version.

Reviewer 1: P6L29: Here you mention again about Bariloche plot (green). Is it the same with previous sentence (Andean lake)?

Reply: We agree with the reviewer in that there is a doubling and the sentence is redundant. We will remove the sentence.

Reviewer 1: P7L8: Rewrite the sentence. Grammatically incorrect.

Reply: We will rephrase it: "Data of Patagonian GNIP stations are in line with or close to the meteoric water line."

[Figure]

Reviewer 1: P7L14-15: Here you mention the $\delta$ difference between Bahia Blanca (BBL) and Bariloche (BAR). But in the next sentence, you mention about temperature difference of 0.5 degree celcius and Figure 2g and 2h. When I look to your Figure 2, then I found BBL is Figure 2g (correct) and BAR is Figure 2e (not correct). The temperature difference is also not correct.

Reply: The reviewer is right that there is a mistake in the submitted text. The comparison refers to Bahia Blanca and Puerto Madryn (not Bariloche). The reference to Fig 2g and 2h therefore is correct. We will rephrase it: "A difference of 5 ‰ between the $\delta$18O of precipitation of Bahia Blanca and Puerto Madryn is, however, not explainable by on average 0.5 °C temperature difference between both sites (Fig. 2 g, h)."

Reviewer 1: P7L15-17: I will say the discrepancy in $\delta$ is due to temperature different or altitude different, which leads to orographic isotope effect.

Reply: The difference between Bahia Blanca and Puerto Madryn is not explainable by temperature or altitudinal effects as both are at approximately the same altitude and have similar temperatures. Therefore, the statements are correct except that we erroneously mentioned Bariloche instead of Puerto Madryn. We regret that this led to confusion.

Reviewer 1: P7L21-27: For Rayleigh distillation method, please see my general comments.

Reply: We will add additional explanations about Rayleigh isotope factionation in a revised version.

Reviewer 1: P8L4-5: Missing location for Madre de Dios and Los Glaciares. Please see my general comments.

Reply: We accept this point of critique and will add the sites in the map (Fig. 1).

Reviewer 1: P8L10: Rewrite the sentence. Grammatically incorrect.

Reply: We will rephrase the sentence in the revised version.

Reviewer 1: P8L19: You may re-write the sentence into: ...., respectively, using the liquid-ice fractionation values taken from Majoube.....

Reply: We will rephrase the sentence: "As all GNIP data used are in line with the GMWL within measuring uncertainties (Fig. 5c), we also refused to use poorly defined local meteoric water lines (LMWL) instead of the GMWL."

Reviewer 1: P9L17: Delete word: of downwind.

Reply: We will delete it in the revised version.

Reviewer 1: P9L19: Replace "almost plots on the GMWL" with " are plotted close to GMWL".

Reply: We will replace it in the revised version.

Reviewer 1: P14L2: missing the: the selected GNIP stations.

Reply: We will add "the" in the revised version.

Reviewer 1: P14L6: You stated GNIP period from 1984-2014. However, Figure 2 d-f say 1970-2000. Which one is the correct one? Can you also please improve the Figure 2? I can hardly read the wind vectors.

Reply: Fig. 2d-f is not from the GNIP dataset; therefore, the time interval is different (1970-2000) and the time interval mentioned in the caption is indeed wrong. In the revised version we will adapt the time interval. We will also generally improve the figure.

Reviewer 1: P17L4: The sentence: for 1984 and 1999, respectively is not correct.

Reply: We will remove the full stop before "for 1984 and 1999, respectively."

Reviewer 1: P18L4: missing "and". Stars represent Patagonian sites and circles represent Pampas......

Reply: We will add "and".

Reviewer 1: P19L2: ....values of lentic waters for selected regions...

Reply: We will replace "of" with "for".

Reviewer 1: P22L2: Change PUW with PDW and change stippled line into dash lines.

Reply: We will change the second "PUW" in the figure caption with "PDW" and "stippled" with "dashed".

Reviewer 1: References Berkelhammer, M., Risi, C., Kurita, N., and Noone, D. C.: The moisture source sequence for the Madden-Julian Oscillation as derived from satellite retrievals of HDO and H2O, J. Geophys. Res. Lett., 117, 2012. Van der Ent, R. J., Savenije, H. H. G., Schaefli, B., and Steele-Dunne, S. C.: Origin and fate of atmospheric moisture over continents, WRR, 46, 2010. Worden, J., Noone, D., Bowman, K., and TES team.: Importance of rain evaporation and continental convection in the tropical water cycle, Nature, 445, 2007. Noone, D.: Pairing measurements of the water vapor isotope ratio with humidity to deduce atmospheric moistening and dehydration in the tropical midtroposphere, J. Clim., 25, 2012.

Reply: For the sake of completeness we will add some of the listed references where appropriate. However, we need to stress here again that the approaches and results of these references are not directly comparable to the results of our study. As a direct comparison is not possible and beyond the scope of our study, we do not intend to exhaustively discuss the satellite-inferred data.

Replies to comments of Reviewer 2:

Reviewer 2: This paper evaluates the stable isotope compositions of precipitation, lentic waters, and lotic waters in that area to characterize and understand isotope fractionation processes associated with orographic rainout, moisture recycling and moisture sources. I think this paper is relevant since it studies isotopes and the relevant literature cited, however it requires major revisions. The manuscript is interesting but

written is poor. It is difficult to catch the authors' messages to readers.

Reply: The critical comments at the end of the paragraph unfortunately are rather unspecific. Thus, it is not possible to directly respond. There may be linguistic flaws in the text, but no example is given by Reviewer 2. The message to the readers will be improved by adding more background details, as suggested by the reviewer in the specific comments.

Reviewer 2: The results are original and represent an important contribution to the understanding of the controls on hydrogen and oxygen isotope composition. However, my main concern is that the problem statement is not clearly defined and that the field description is not sufficient as it is.

Reply: We will clarify the information on the research objectives and the field description in a revised version.

Reviewer 2: Specific comments: Abstract: This is mostly composed of the Introduction and Material and methods. So, I do not think the abstract is attractive.

Reply: We agree that our results and conclusions should be highlighted in more detail in the abstract. We will do so in a revised version.

Reviewer 2: Introduction: Describe more paper about the stable isotope compositions! which problems? why is this study important?

Reply: We agree that this part could be more elaborated and will add additional information in a revised version.

Reviewer 2: Material and methods: Detail more the monitoring and describe the collection methodology.

Reply: We can add additional methodological details in a revised version.

Reviewer 2: I think the paper would be better if the authors combined the results and discussion.

Reply: We do not see a reason in the reviewer's comment for merging results and discussion to one chapter and prefer the chapter structure as it is.